# The geography of intergenerational social mobility in Britain

Paul A. Longley 🅾 [1✉], Justin van Dijk 🅾 [1] & Tian Lan 🅾 [1✉]

Empirical analysis of social mobility is typically framed by outcomes recorded for only a single, recent generation, ignoring intergenerational preconditions and historical conferment of opportunity. We use the detailed geography of relative deprivation (hardship) to demonstrate that different family groups today experience different intergenerational outcomes and that there is a distinct Great Britain-wide geography to these inequalities. We trace the evolution of these inequalities back in time by coupling family group level data for the entire Victorian population with a present day population-wide consumer register. Further geographical linkage to neighbourhood deprivation data allows us to chart the different social mobility outcomes experienced by every one of the 13,378 long-established family groups. We identify clear and enduring regional divides in England and Scotland. In substantive terms, use of family names and new historical digital census resources are central to recognising that geography is pivotal to understanding intergenerational inequalities.

[1] Department of Geography, University College London, Gower Street, London WC1E 6BT, UK. ✉email: p.longley@ucl.ac.uk; tian.t.lan@ucl.ac.uk

Social mobility has been defined from a political and policy perspective as 'the link between a person's occupation or income and the occupation or income of their parents', with stronger links identifying lower levels[1]. Empirical research into the concept, its occurrence and its policy implications has deployed novel methods to link data that compare the income and other socioeconomic characteristics of parents and their children, sometimes as a precursor to evaluation in the context of the present-day local employment opportunities available in local administrative jurisdictions. This is nevertheless limiting for several reasons: First, heavy reliance upon income and occupational characteristics precludes permanent income measures, life-course asset accumulation and intergenerational transfer of family wealth[2]. Disposable income is notoriously difficult to define and eligibility for social welfare payments (such as child free school meals) is discrete rather than graduated. Second, relative life chances are often cumulative and divergent outcomes of circumstances are transmitted through multiple, rather than just two, generations[3]. Although some longer term intergenerational mobility investigations have recently emerged following provision of historical microdata[4], most studies remain limited by their small sampling fractions of relevant population. Third, family (as opposed to individual) characteristics may systematically enhance intergenerational transmission of human capital between group members, exemplified by Clark and Cummins[5] analysis of social mobility outcomes for family groups with or without forbears that attended England's ancient universities (Oxford and Cambridge) in medieval times. Fourth, socioeconomic milieux, present day and historical, shape social capital formation while functional regions govern access to economic opportunities and ways of working[6–8]. Fifth, the effects of economic migration, recent or historical, upon observed inequalities of outcomes are ignored. Sixth, outcomes for hard-to-reach groups such as workers in the informal economy or migrants may not be detected.

The Great Britain-wide Index of Multiple Deprivation[9] (IMD) provides a summary measure of the relative degree of hardship experienced by the residents of every neighbourhood area (typical population 1500 and termed Lower layer Super Output Area, LSOA, in England and Wales). Its dimensions are strongly indicative of social mobility outcomes, comprising weighted measures of income, employment, health, education, crime, barriers to housing and living environment[9]. These measures together summarise physical and social conditions that are less transitory than individual or household income and occupational characteristics alone. The precise make-up of this composite measure differs between UK countries, as does the precise date of compilation (2019 for England and Wales, 2020 for Scotland) but percentile scores can be considered broadly equivalent between neighbourhoods throughout Great Britain. The GB-wide distribution of scores is highly variegated, with the functional regions surrounding urban centres showing mixes of areas of greater or lesser hardship and no GB-wide patterns (see Supplementary Fig. 1). Many households filter upwards to successively less deprived neighbourhoods during the household lifecycle, consistent with filtering theories of neighbourhood dynamics (see, for example, ref. [2,10]).

Whilst there is by no means any perfect correspondence between family name and kinship, and common names such as Smith or Brown are unlikely to signify common ancestry, the vast majority of family names are known to be regionally or locally concentrated around the locales in which they were first coined between the 12th and 14th centuries[11]. Regional and local concentrations of family names are very much the norm in the historical period and these distributions endure today. Illustrative summary regional distributions for the names Offer and McPhee are shown in Fig. 1a, b, respectively, using kernel density

estimation for presentation purposes (see ref. [12]). Most common names such as Smith are not regionally distributed and mirror the population distribution across Great Britain (Fig. 1c). Comparison of such maps makes it possible to chart the contagious and hierarchical diffusion of every long-established family name through the British settlement system.

While acknowledging that it is a misnomer to describe all Smiths or Browns as constituting a 'family group', we argue that the term is appropriate for the bearers of most all other family names—albeit that it connotes shared geographic origins more strongly than common ancestry. Clark and Cummins[13] have demonstrated that membership of specific family groups, as indicated by family names, is associated with distinctive intergenerational social outcomes, but this phenomenon has never been investigated for entire populations or related to the detailed geographic context in which particular family names are concentrated. Cheshire and Longley[11] and Kandt et al.[6] have documented the tendencies for geographic concentrations of most British family names to endure between generations, and for them to diffuse over time to areas adjacent to their ancestral heartlands (contagious diffusion) or cascade through high to successively lower order urban areas in the settlement system (hierarchical diffusion). This may result from residential mobility during the household lifecycle or migration in response to differentials in economic opportunity[14]. Until at least the current generation, family names have usually been passed down the male line, but the historical tendency for marriages to be formed between local partners[15] means that issues of gender do not exert a significant impact upon regional concentrations or over-all geographic patterns.

As such, most family names provide universal if under-utilised markers of geographic ancestry, notwithstanding that common and geographically widespread family names dampen over-all regional patterning amongst the population at large. Examination of precisely georeferenced family name distributions, present day and historical, alongside IMD scores offers the potential to progress study of intergenerational social mobility since, with respect to each of the six shortcomings listed above. First, the IMD provides a wider ranging and indelible summary measure of living standards by which to evaluate outcomes of social mobility. Second, the roots to intergenerational inequality can be traced to any period for which family names are available and geographical analysis of inequalities of outcome can be conducted. Third, many of the effects of family group membership can be accommodated over intergenerational timeframes. Fourth, geographic context ('place') can be accommodated at any convenient scale of analysis. Fifth, the effects of migration, or the lack of it, upon family group outcomes can be measured. Sixth, it is possible to frame analysis in terms of entire relevant populations—such as overwhelmingly 'long-settled' family groups—and limit or control extraneous considerations such as the wider effects of international migration upon observed outcomes.

In this work, we seek to extend empirical social mobility analysis through truly intergenerational, population-wide analysis of the family groups that make up the entire long-settled population of Great Britain. We do this by tracing the geographic origins of the family groups present in nineteenth-century Britain and then profiling the neighbourhood environments in which their present-day descendants reside. We relate observed present-day levels of hardship, as measured using the harmonised IMD, to family group membership and geographic location. We do this by calculating family group average IMD scores and examining their correspondence with (a) present-day neighbourhood type and (b) residential locations of nineteenth-century ancestors. The null hypothesis underpinning this analysis is that family names taken at some point between the 12th and 14th centuries bear no

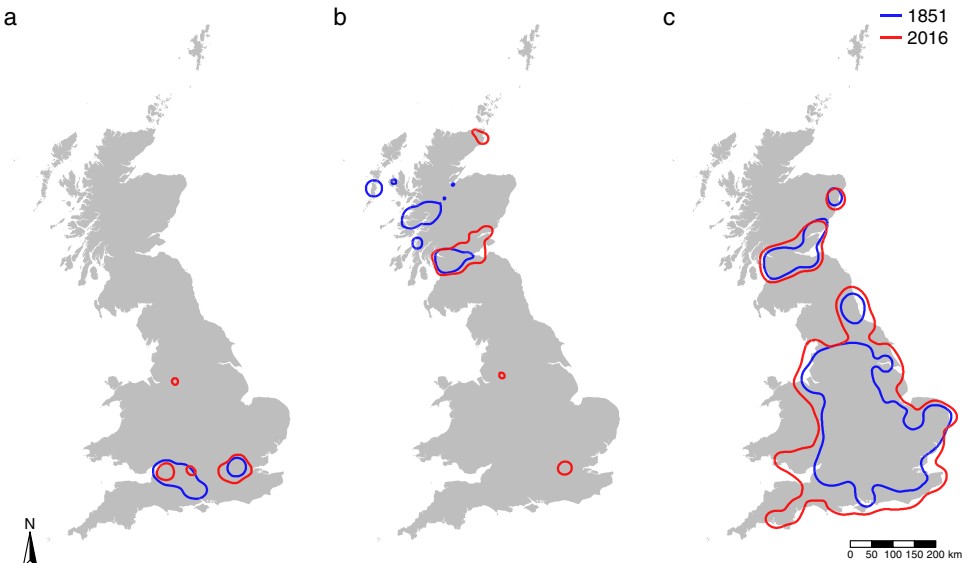

**Fig. 1 Surname concentrations.** Kernel density estimates (KDEs) of the 1851 and 2016 population-weighted geographic distributions of bearers of the family names (**a**) Offer; (**b**) McPhee; and (**c**) Smith. Regional names manifest urban and inter-urban migration over time, while very common names mirror the GB-wide population distribution. (Note: KDEs provide measures of relative density, weighted by the general population density. They are range standardised to improve comparability using the method detailed in ref. [12]).

correspondence with observed patterns of hardship today. The universal coverage of common family names such as Smith (Fig. 1c) is clearly consistent with this, and such distributions will self-evidently dampen any effects attributable to names that remain locally or regionally concentrated. The alternative hypothesis is that some combination of family fortunes and geographic location conspire to create present-day outcomes. Throughout, this analysis assumes that rare family names in 1851 or those subsequently imported from abroad are exogenous to local levels of hardship: we return to this in our discussion. Bearers of names not classified as 'long established' account for 36.7% of the 50.1 million adults recorded in the 2016 consumer register.

## Results

We define the outcome of intergenerational social mobility in terms of individual experience of neighbourhood deprivation, $D$. The average level of deprivation $\bar{D}$, experienced by the $N_j$ members of every long-established family group $j$ across a set of $K$ neighbourhoods is defined as

$$\bar{D}_j = \sum_{k=1}^{K} N_{jk} D_k / N_j \qquad (1)$$

With caveats, $\bar{D}_j$ is taken to manifest the intergenerational social mobility outcome of family group $j$ by assuming its direct correspondence with the average quality of neighbourhoods in which family group members live. We predict the quality of any neighbourhood $k$, $\hat{D}_k$ as the averaged deprivation score of its $N$ residents drawn from any of the $J$ long settled family groups.

$$\hat{D}_k = \sum_{j=1}^{J} N_{jk} \bar{D}_j / N_k \qquad (2)$$

Differences between observed and predicted deprivation scores manifest geographically local social mobility outcomes—positive where residents have exceeded their family group's social mobility outcome, and negative where they have fallen short of it. We can better envisage the genesis to any patterning in under- and over-achievement for any convenient historical time period $y$ by

examining the future deprivation that an area's historical mix of residents will bequeath upon its descendants:

$$\hat{D}_{k,(t-y)} = \sum_{j=1}^{J} N_{jk,(t-y)} \bar{D}_{j,t} / N_{k,(t-y)} \qquad (3)$$

In a world with no intergenerational frictions on social mobility, any geographic variation in $\hat{D}_k$ would be local and attributable to residential filtering of, for example, affluent later life-stage households. There would likely be little local patterning in distant historical periods given shorter lifespans and limited residential mobility. To anticipate our empirical results via coupling population data for the 1851 census with the present day, $\hat{D}_{k,1851}$ turns out to have a very strong and regional geography, while that of present day (2016) $\hat{D}_{k,2016}$ reveals a less stark but distinctive regional pattern in which population migration has mitigated some of the structural intergenerational inequalities. We suggest that historical connectedness of ancestral family residence to London is an important predictor of the inter-generational variation in $\bar{D}_j$ that is observed today.

**Present-day family group inequalities of outcome.** A fundamental premise to our analysis is that neighbourhood quality, as measured by IMD, manifests the disposable and permanent incomes of residents. Under our null hypothesis we expect no relationship between family group IMD and neighbourhood quality. Precise georeferencing of every adult included in the 2016 consumer register makes it possible to test this hypothesis by assigning (neighbourhood scale) Lower layer Super Output Area (LSOA) harmonised IMD scores to each adult resident and then averaging these scores for each family group.

Figure 2a shows that the mean values of the 13,378 long-established family names are far from uniform and are instead associated with sharply different neighbourhood conditions spread across the 40–60 range. The grand mean value of 52.89 for these long-established family groups is above the GB zonal average (that for Smith is 50.65) but the mean scores for 23.84% of names are below 50. Figure 2b provides an extract from the population distribution, including the family names that are

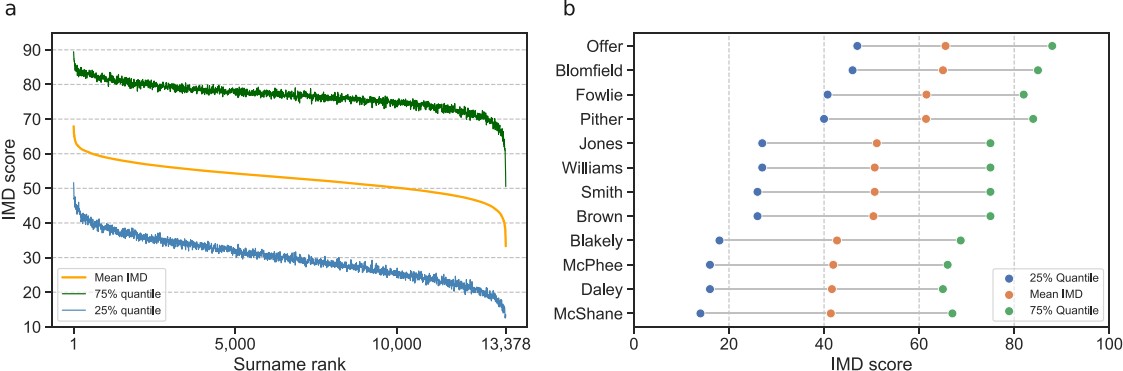

**Fig. 2 Mean Index of Multiple Deprivation (IMD) scores distribution of family names. a** Mean IMD scores and inter-quartile ranges for the 13,378 long-established family names with 100 or more bearers in 1851 and 200 or more adult bearers in 2016, ranked from least to most deprived; **b** names extracted from the least deprived, middle and most deprived parts of the distribution. Bars display the interquartile ranges (IQRs) about the mean IMD values. The numbers of adult name bearers (n) in 2016 in (**b**) are 677 (Offer), 513 (Blomfield), 524 (Fowlie), 613 (Pither), 424,930 (Jones), 295,271 (Williams), 550,878 (Smith), 260,944 (Brown), 562 (Blakely), 3756 (McPhee), 7096 (Daley), and 3079 (McShane).

mapped in Fig. 1. The observation that family group membership bears an identifiable correspondence with intergenerational inequalities is consistent with historical research establishing that specific family groups generate, sustain and transmit high levels of human and social capital through the generations[13]. The evidence of Fig. 2a is that these differences permeate all family echelons—seemingly regardless of any requirement of common ancestry. The inter-quartile ranges indicate remarkably constant variation in outcomes between family group scores irrespective of numbers of bearers and their symmetry about the mean, indicates similar ranges of outcomes for each family group. Our contention is that the population-wide variability in mean values arises in significant part from the historical geographies of different family groups. Our exploratory analysis of the names making up the upper and lower tails of the distribution suggests they predominantly comprise distinctive locally concentrated names with rather few bearers, but our primary focus is to examine over-all population characteristics.

Our expectation is that family groups that are relatively deprived will tend to live in highly deprived neighbourhoods, and vice-versa. 'Family group IMD' percentile scores ($\bar{D}$ in Eq. (1)) are averaged across all adult members of long-established family groups in each neighbourhood (LSOA), to estimate $\hat{D}_k$ in Eq. (2). The averaged LSOA values shown in Fig. 3 present the combined effects of scores attributable to family group members who have either migrated or remained in their ancestral areas. They are overwhelmingly above the grand mean value of 50.43, although there is a general north–south divide across England and Wales and the least deprived scores in Scotland lie north of its central lowlands. The lowest scores are concentrated in the cores of metropolitan areas, particularly in northern England and southern Scotland. OLS regression of family group IMD scores against the observed harmonised IMD scores yields a significant positive relationship (adjusted $R^2 = 0.144$, $p = 0.000$, intercept = −630.515, coefficient = 13.200).

**Intergenerational patterns of inequality**. We next assign each individual in the 1851 Census their family group 'future IMD' score. This indicates the combined human and social capital that each individual would, on average, bestow upon their descendants to be crystallised in the neighbourhoods in which their descendants would live. These are then summed and averaged for every harmonised historical parish in order to measure the future neighbourhood trajectories that each parish would bestow upon descendants ($\hat{D}_{k,(t-y)}$ in Eq. (3)).

The mapped results (Fig. 4) reveal a very clear 'north–south divide' in England, bounded by Devon in the west and a line extending north-eastwards between the Severn Estuary and the Wash. An east–west gradient is also apparent in Scotland, with eastern areas sharing similarly high levels of future hardship to the nineteenth-century industrial areas of Liverpool, Manchester, Northumberland and the Fylde Coast. Within the southern region, the scores associated with London are below average, reflecting the full mix of long-established names already found there in 1851. Figure 4 presents a starker and more divergent GB-wide pattern than Fig. 3 and demonstrates the strong concentration of favourable intergenerational opportunity in the south and east of England and Wales. Further analysis (see Supplementary Figure 2) confirms that these patterns endure when the future IMD scores are calculated using 1861, 1881, 1891, and 1901 historical Census data from the I-CeM collection.

Following harmonisation of 1851 parish geography with contemporary LSOAs, we regress the latter against the former. The results indicate a very weak but significant relationship ($R^2 = 0.034$, $p = 0.000$, intercept = −89.117, coefficient = 2.802). There is considerable spatial dependency in the residuals, and geographically weighted regression[16] (GWR) with adaptive bandwidths produces a pseudo-$R^2$ of 0.548 and a stronger relationship (mean standardised intercept = 0.035, mean standardised coefficient = 0.128).

**Lifetime migrants living in London and future IMD outcomes**. The transition between historical Fig. 4 and the attenuated present-day disparities of Fig. 3 was likely achieved through net north–south migration as documented, inter alia, by Schürer and Day[17]. In this respect the enduring historical pattern for England and Wales bears close correspondence with Schürer and Day's (ibid: Fig. 7) mapping of the places of birth of first-generation migrants to London, expressed as proportions of all births in the origin parishes as recorded in the 1851 Census for England and Wales. (The Scottish Census did not record parish of birth). Although this measure is asynchronous (the numerator of first-generation parish migrants by definition relates to different individuals than the denominator count of births), this can be considered a summary indicator of connectivity to the capital, measured as an outcome of awareness and opportunity to migrate.

Establishing a statistical relationship between historical migration flows and present-day hardship (IMD) requires matching of the historical pattern of parish boundaries with present-day LSOAs: the I-CeM standardised birth parishes are not linked to either of the two available sets of consistent parishes. We therefore create a best fit lookup between the two zonal schemes by matching the birth

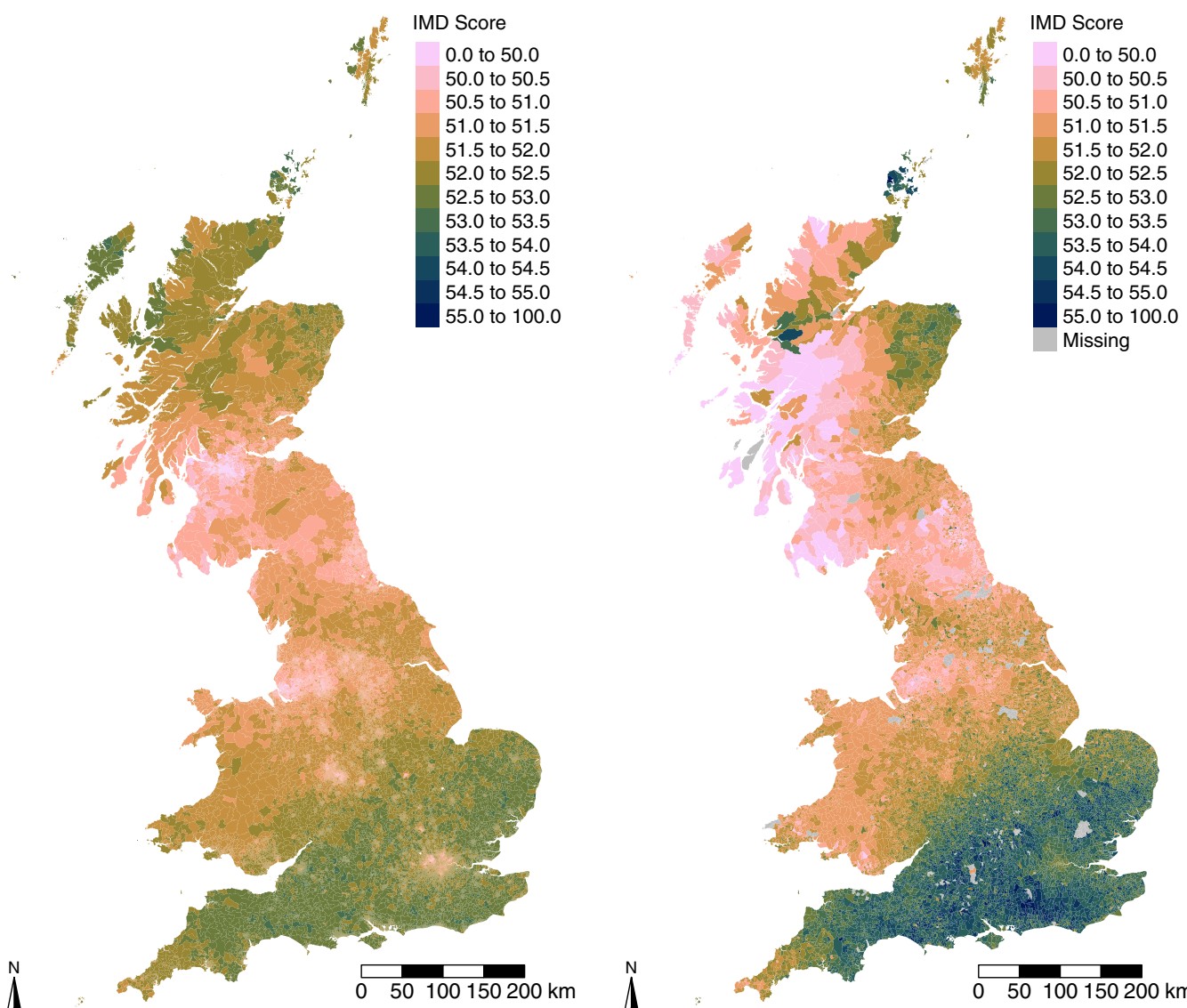

**Fig. 3 Modelled Index of Multiple Deprivation (IMD) scores.** 2016 modelled LSOA IMD percentile scores, calculated as the average 'family IMD scores' of all residents drawn from long-established family names. There is a distinctive pattern in which the southwest of England has on average a higher modelled IMD score than the rest of Great Britain.

**Fig. 4 Future Index of Multiple Deprivation (IMD) scores.** 1851 modelled consistent parish IMD percentile scores, calculated as the average 'family IMD scores' of all residents from long-established family names. The map reveals a pronounced 'north–south divide' between the southwest of England and the rest of Great Britain.

parish strings to the names of the consistent parishes. The full assignment process results in a list of the total number of individuals that were born in each English/Welsh consistent parish and were living in London in 1851, along with the total numbers of births in each of these areas in 1851 (see Fig. 5).

Using the harmonised parishes of England and Wales as units of analysis, OLS regression of observed IMD against the proportion of lifetime parish migrants living in London yields a very weak, but significant, relationship ($R^2 = 0.004$, $p = 0.000$, intercept = 56.787, coefficient = 0.202). There is evident spatial dependency in the residuals, and multiband GWR results identify a moderately strong relationship with a pseudo-$R^2$ of 0.564 (mean standardised intercept = 1296.764, mean standardised coefficient = 1395.247).

## Discussion

Although recognised as a one of the most important challenges across social science, geographers have not developed a clear

theoretical literature on the sources and operation of inter-generational social mobility, in contrast for example to those of economists (e.g. Becker and Tomes[18] and Chetty et al.[19]). The analysis reported here establishes the nature and extent of spatial equity issues when considering intergenerational (as opposed to generational—see ref. [20]) social mobility. We have developed a population-wide analysis of intergenerational social mobility across the entirety of Great Britain, spanning 1851-2016. Our results demonstrate that inequality is fundamentally inter-generational and substantially geographic. This has implications for social mobility studies of all advanced economies, since it is demonstrably insufficient to benchmark change over just one or two generations. There has been no level playing field for any recent British generation, and the geography of ancestry should be considered alongside other processes of human and social capital formation, including education, household structure and labour market policy. Our results also demonstrate that structural regional divides in social mobility have only been partially offset

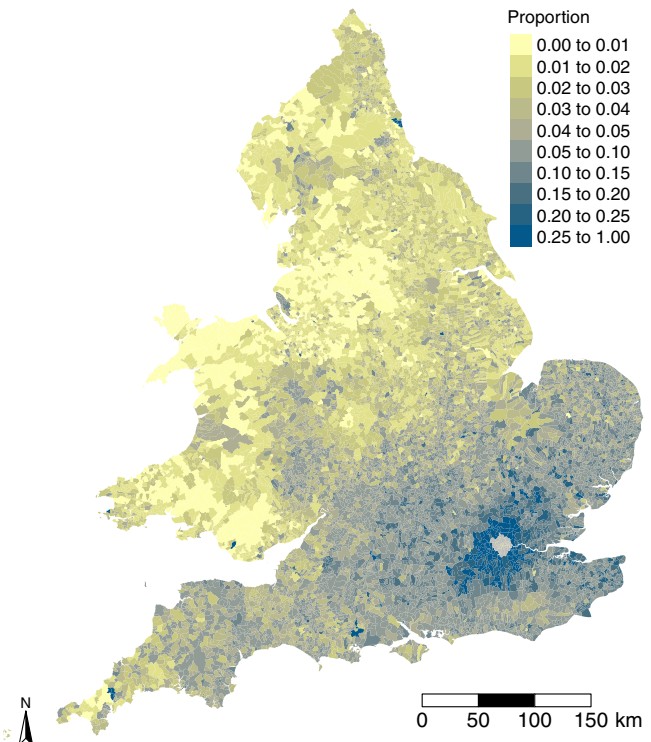

**Fig. 5 Lifetime migrants living in London.** 1851 lifetime migrants living in London by English/Welsh registration district as a proportion of those born in that place (after Schürer and Day[17]: Fig. 7).

by the effects of migration. New methods and data sources are needed to comprehensively link micro level geographies and to address issues of scale, aggregation and replicability in geographic analysis.

The foremost finding of this paper is that today's social mobility outcomes for long-established British family groups are deep-rooted in intergenerational, population-wide inequalities of opportunity. These are evident in family group deprivation scores that, when aggregated, show very evident geographic patterning. In England this enduring structural pattern is driven by the historical functional linkage of places to London. Family roots in northern industrial cities remain associated with unfavourable outcomes today, while an east–west gradient in Scotland may reflect the enduring dearth of opportunities bequeathed upon the descendants of migrants from Ireland. Our population-wide analysis identifies a structural historical geography component to deprivation (Fig. 4), while mapping its distribution today (Fig. 3) confirms its mitigation because of migration and mixing of the population that has taken place over the last 165 years. Migration has mitigated but not eliminated the structural inequality that is attributable to geography—indeed, if migration differentially selects individuals with the greatest skills and abilities it may compound geographic effects in ways not readily detectable using family names analysis. Regardless of this, it is apparent that government regional policies have not fully compensated for the disadvantages conferred by geographic location.

The message that inequality is deep-seated, intergenerational and substantially geographic should be accommodated into social mobility analysis in Britain. This finding likely has implications for studies of all advanced economies. It is not sufficient to benchmark change over only the last two generations. Simply put, there has been no level playing field for any recent generation, and the geography of ancestry should be considered alongside other processes of human and social capital formation.

High-quality neighbourhoods that are scattered throughout Great Britain today disproportionately house residents whose ancestors were on the right side of historical divides, and vice versa.

This important finding is only made possible by population-wide spatial analysis of the historical geography of family names. These universal tokens of geographic identity have been under-utilised in geographic research. The results presented here are clear and unequivocal, driven by breadth of coverage and data intensive georeferencing. The clarity of our substantive conclusions is perhaps all the more remarkable because of a number of caveats to family names analysis. First, the assumption that names overwhelmingly inherited through the male line does not create bias, consistent with Pooley and Turnbull's observation[15] that women historically take spouses from the same localities: social selectivity has nevertheless diminished some names potentially deemed awkward: for example, the 992 bearers of the name Smellie in 1851 had reduced to 366 adult bearers in 2016. Second, we do not attempt to accommodate any relationship between high apparent fertility of some family groups or their likely supplementation by post 1851 immigration from Ireland: the 100 Mannions in 1851 had grown to 3275 adults by 2016, for example. Third, the names identifying some long-settled family groups are polycentric in origin, which may attest to homonyms with very different social mobility connotations or overseas immigration (particularly from Ireland). Fourth, historical naming conventions have been spatially heterogeneous, perhaps leading to coarser granularity of future IMD geographies in Wales, for example, where many individuals share a relatively small range of family names. Fifth, although the overwhelming majority of family names have local or regional connotations, many very common ones, such as Smith (280,277 occurrences in 1851 and 550,878 adult occurrences in 2016) or Brown (133,633 occurrences in 1851 and 260,944 adult occurrences in 2016), do not. Names that are both common and widespread have mean IMD scores very close to the grand mean of 50.43 of all adult residents recorded in the 2016 consumer register and exert a dampening effect upon the historical parish average scores. Finally, sixth, exclusion of names with fewer than 100 bearers in 1851 and 200 adult bearers in 2016 is a precaution to avoid transcription errors but may exclude small family groups with distinctive future IMD profiles.

These caveats and qualifications point to some important extensions to the analysis presented here. In defining our population of interest as 'long-settled family groups', we bring focus to the intergenerational inequalities that the British state has bestowed and sustains today. Most observers would concur that the state is less culpable for the average neighbourhood deprivation scores of groups recently arrived from overseas: yet 'recent' is a relative term, and it is for further analysis to identify and characterise the social mobility of migrants that have entered British society during the twentieth and twenty first centuries. Social mobility outcomes for descendants of Irish immigrants are an important case in point, since Great Britain has been a destination for Irish migrants throughout recorded history, and particularly in the Irish famine years prior to the 1851 Census. Our own exploratory analysis of 1851 Census records of Irish birthplaces of individuals resident in Great Britain in 1851 identifies no clear cut-off point beyond which a family name could be deemed 'Irish' and hence imported from abroad, but analysis of the future IMD scores of names with substantially recent Irish origins could provide a measure of the mobility of a long-established but disadvantaged migrant group—not least in Scotland, for example, where circa 7 per cent of the population in 1851 comprised generational migrants from Ireland. Such analysis would require development of heuristics to differentiate between longer established and more recently arrived migrants.

The data infrastructure created here might also be developed in order to examine relative concentrations of local and non-local family groups in neighbourhoods characterised by high or low levels of deprivation, with the expectation that low proportion neighbourhoods will disproportionately comprise successful migrant bearers of non-local names. This hypothesis presumes that, in general, human and social capital is passed down through the generations, cognisant of attrition of this effect across multiple generations[18]. It also poses challenging questions of how 'local' is defined in regional context, in relation to spatial interaction patterns[21], geographies of educational attainment or labour market functioning[22] and 'surname regions'[6].

Social mobility outcomes bear strong correspondence not only with neighbourhood choices but also with consumption of other goods and services. Deprivation analysis is part of the field of geodemographics, or the analysis of people by where they live[23]. This field has a rich history of investigating characteristic patterning of types of social area across different urban areas, consistent with urban system-wide processes such as suburbanisation, residential filtering or gentrification. As such, geodemographics is essentially nomothetic social science[24], bringing focus to classification of the underpinning social processes that repeat across entire settlement systems. Use of georeferenced data grounded at the level of the human individual makes possible the flexible adaptation between the present-day and historical areal units used here. If inferences can be successfully drawn at the level of the individual (e.g. inference of ethnicity or recency of migration from family names) issues of ecological fallacy, scale and aggregation are circumvented.

The message of this paper is that social mobility is historically and geographically contingent, hence unique. Social patterns of high and low IMD within functional regions may be highly variegated, yet repetitive. But the intergenerational analysis developed here demonstrates that different places can provide quite different platforms for intergenerational processes and that the geodemographic landscape is underpinned by unique and distinctive historical patterns of social relations. Defining what is intrinsically unique and what is repetitive has implications for the specification, estimation and testing of geodemographic representations, as well as their replicability[24]. New digital data infrastructure can now couple historical censuses with contemporary consumer registers at the level of the georeferenced human individual. This provides a framework for charting evolving systematic features of these unique human landscapes, opening up the prospect of developing geodemographics as a blend of idiographic as well as nomothetic approaches to social and spatial mobility over intergenerational timeframes.

## Methods

**Historical census microdata and consumer registers**. The approach developed here would not have been possible prior to the advent of digital historical records[25,26] and the development of consumer registers of the names and addresses of most contemporary adult residents[27]. The 1851 Census microdata of Great Britain is one of the Integrated Census Microdata[25] (I-CeM) collections, provided by the UK Data Service under special licensing arrangements. There were 20,610,325 residents enumerated in Britain in 1851. The total numbers of records and data coverage for all of the Censuses are summarised in Supplementary Table 1. The digitised and standardised census microdata include individual level information about every resident's name, address, standardised birthplace, harmonise parish, and other socioeconomic variables. Southall[28,29] describes the creation of digital boundaries that harmonise historical parish extents for 1851–91 and 1901–11. These digitised consistent parish boundaries are used to georeference individual census records from the 1851 Census for bearers of every family name with 100 or more bearers.

In order to facilitate comparison with present-day family name distributions, we also filter out occurrences of every family name with fewer than 200 bearers in the 2016 consumer register. The 2016 consumer register is part of the Linked Consumer Registers[27] (LCRs) and comprises of the names and addresses of a very large proportion of all the adults residing in the United Kingdom. The main source of data feeding into the LCRs are the public versions of the GB Electoral Register. This is supplemented with consumer data from 2002-2016 onwards to capture

those individuals that opt-out of the public version of the electoral register or are not eligible to vote. All component registers are precisely georeferenced by matching address records to the Ordnance Survey (GB) AddressBase Premium product. The LCRs pertain to the vast majority of the adult population in the UK and have been used in demographic research on topics like ethnic segregation (see ref. [30]) and residential mobility (see ref. [31]).

**Birth parish matching**. To chart the lifetime migrants living in London in 1851, we create a best fit lookup by matching individual record birth parish strings to the names of the consistent parishes This first requires hierarchical string searches to match the I-CeM standardised birth parishes to consistent registration district, registration subdistrict, or parish string within the corresponding birth county. For all birth parishes for which we cannot identify a corresponding consistent parish, we create a proportional lookup; i.e. any individual that we can assign only to a birth county is assigned as fractions to all candidate birth parishes in the county. In a second step, we calculate for each English/Welsh birth parish the number of migrants living in London in 1851. For this we define London as all the parishes intersecting with the total connected road network of London in 1851 (see ref. [32]). In the digitised and enhanced 1851 Census of Population 18,615 unique birth parish and birth county combinations are found: 12,353 of these combinations could be assigned to one of the digitised consistent parishes (see ref. [28,29]). For the remaining 6262 of these birth parish and birth county combinations, we used proportional assignment to a registration county (4591), a registration sub-district (1086) or a registration district (498). 87 unique combinations could not be assigned. The majority of these 87 combinations were recorded in the 1851 Census as "not applicable", "not coded" or "unknown". The 12,353 unique combinations that were matched at the parish level account for 9,006,020 out of the 17,153,781 individuals that bear a surname in 1851 that we considered 'long-established' in our analysis.

These highly labour-intensive endeavours make it possible to compare the precise geographic distributions of all bearers of significant 'long-established' family names in Great Britain in 1851 with the present-day distributions of adult bearers.

**Reporting summary**. Further information on research design is available in the Nature Research Reporting Summary linked to this article.

## Data availability

The 2016 consumer register data (https://data.cdrc.ac.uk/dataset/linked-consumer-registers) and the I-CeM historical census data 1851–1911 (https://doi.org/10.5255/UKDA-SN-7856-2) used in this research are available upon successful application to the Consumer Data Research Centre (CDRC: cdrc.ac.uk) and the UK Data Service (UKDS: ukdataservice.ac.uk), respectively. The analysis conducted here was undertaken under special licensing arrangements for access to individual level data. As such, the data are only available upon successful application to these Economic and Social Research Council-funded data centres.

## Code availability

All data analyses and computations are conducted with the standard R and Python packages. R version 4.1.0 was used for the analysis, using the data.table (1.14.0), raster (3.4.13), scico (1.2.0), sf (1.0.2), sparr (2.2.15), tidyverse (1.3.1), tmap (3.3.2) and tmaptools (3.3.1) packages. The birth parish to consistent parish string matching process was executed in Python (3.8.3) using the pandas (1.0.4), numpy (1.18.5), and rapidfuzz (0.9.1) libraries.

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

## Acknowledgements
This work was supported by the Engineering and Physical Sciences Research Council [EP/M023583/1]; Economic and Social Research Council [ES/L011840/1].

## Author contributions
P.A.L., J.V.D., and T.L. conceptualised the study, designed the research, conducted the analyses and wrote the manuscript. All authors read and approved the final version of the manuscript.

## Competing interests
The authors declare no competing interests.
