## [Peer Review File · Nature Communications]

Reviewer comments, first round:

-

Reviewer #1 (Remarks to the Author):

Thank-you for submitting this interesting & idea provoking piece of work (see below!).

I encourage the authors to take a step back from the whole and see whether a different series of subsections might just help the what we are doing, how and what we have found elements.

Rather than a standard Data & Methods and Results sections, I need to declare that I really do like, for successive subsections of work like this, that people say for each part what the resources were, what processing happened and what was found. This is important in work like this whereby one can forget what on earth it was the people did (in methods) to thereby understand the results. To a large extent you do this, but some better divisions and flagging / pinpointing the story would help.

Here, you are a little inconsistent. For example, the introduction moves into data source (the historic bits) but misses the consumer register which, as far as I can see, is not given sufficient explanation. Then, we get the measures being used (lines 111 onwards) and we move into some results while still in the intro.

I'd suggest the intro does what it should by having the social mobility elements and roughly then up to line 72. Then explain (if you can be encouraged to agree with me!) that there will be a set of analyses in which the sources, methods and findings will be self-contained. each of these then says what it will do.

Referees are often asked if something can be omitted and still have a good paper. I find the London subsection a bit of an outlier; not least because it doesn't fit with the GB coverage. I think this would standalone in another paper & maybe you could incorporate the mirror. i.e. names born elsewhere who end up in London.

A few details:

- * As noted, some detail about the consumer register
- * Better colour distinction in fig 1 (I can't see the light blue
- * It looks like the adjusted IMD is between 0 and 100 (and you say range standardisation somewhere later?)
- * Can fig 2 have an everyone else's average (IQRs) to see how the families of interest vary from others?

A couple of ideas for future work:

What is the old and new geography of place name surnames?

What is the old and new geography re IMD of occupation names?

Good luck with all this which is really intriguing work!

Paul Norman

Reviewer #2 (Remarks to the Author):

This paper uses the Britain-wide Index of Multiple Deprivation (IMD) from 2019/20 in combination with the 1851 census and the 2016 consumer register to examine social mobility over a period 170 years. Individuals are assigned to a "long established" family by their last name in 2016 and in 1851, assuming that individuals with the same (uncommon – nevertheless accounting for 63% of the population in 2016) last name stem from the same lineage. Using geo-references in the consumer index, the authors then calculate average deprivation scores for each family, predicted deprivation scores of each neighbourhood and future deprivation scores of parishes in 1851.

The method of using last names is not new and the authors fully acknowledge this. What is new, is the combination with population-wide modern data including geo-references which allows an examination of the geography in social mobility.

The method is sound and the results are novel and interesting. The main contribution, demonstrating the geography of social mobility, will be useful for other researchers but also has policy-relevant implications.

My main comment concerns how the results are put into perspective. The discussion of the main contribution of the paper could be more focused throughout the paper. This would not only improve readability but also bring out more clearly what we actually learn from this exercise (also in terms of context-related implications as opposed to data-related implications).

The discussion on p. 11 line 384-407 summarizes the main contribution of this paper nicely: social processes exhibit geographical persistence. This is a very interesting and policy-relevant result. I would suggest the authors to focus on this and outline more clearly the implications of this demonstration. As it stands now, the authors discuss different motivations/implications not all of which can really be derived from the results. For example, the authors state several times throughout the paper (especially in the discussion) that the results demonstrate that it is not sufficient to look at social mobility over only two generations. This is true, however especially in recent years, with the availability of large historical datasets, the focus has already moved towards long(er)-run social mobility measures over several generations and the authors also cite the literature. Also, at times the paper is presented as showing how geography affects social mobility (e.g. p. 6 line 196: "Our exploratory analysis of the names.... Suggests the influence of distinctive local factors..."). We cannot derive this from the results, it may also be individuals/families exhibiting a certain (im)mobility pattern (or their wealth and land possessions – or lack thereof) leading to distinctive local factors. What we can derive is the persistence in the geographic pattern in social mobility – not the mechanism.

The analysis is done on 13,378 long established family groups. While these cover the majority of the population in 2019, their mobility patterns may be different from common namers, especially historically and in terms of geographical mobility. Also, populations are not actually "linked" but rather classified into family groups based on family names. As such, the statement that the analysis is "truly intergenerational" (p.3 line 54) is somewhat overstated, also given the fact that the deprivation index is not an individual measure. This is unnecessary given the immense datawork of the paper.

The discussion includes a paragraph about immigration, especially from Ireland. The period after 1851 was, however, also a period of massive outmigration. Emigrants were on average poorer and less educated and emigrated in search of better opportunities. This also implies that those who stayed were those who deemed their opportunities at home sufficiently preferable to emigrating.

The illustrations (figures and maps) are very well chosen and illustrative. Especially figure 3, however, calls for a comparison between the modelled and the observed IMD. I understand the authors did not include the observed IMD as this is not the contribution of the paper, but it could be useful for the reader to see the difference between modelled and observed, for example.

Some more details on the methodology (especially the "linking" procedure, incl. match rates) and the different datasets (summary statistics, observation numbers, etc.) would be useful for readers who are interested in the supplementary material.

Reviewer #3 (Remarks to the Author):

Thank you for the opportunity to review this paper, I very much like it. It takes advantage of digitised historical records in order to present the first analysis of inter-generational social mobility for the Victorian population in GB to the present-day population. Area-level deprivation is used to

measure social mobility outcomes for 13,378 'long-established' family groups, exploring how geography of ancestry relates to apparent differences in opportunity.

The key finding emphasises the importance of context, and that – as some may argue is already known – 'inequality is deep-seated, inter-generational and substantially geographic'. Though perhaps not as novel a finding as the paper feels like it implies, it is certainly a fresh and valuable set of evidence giving more substance to what is perhaps already articulated in wider sociological and economic literatures. What is most compelling about this paper is the arguments around under-use of 'universal tokens of geographic identity', and the way in which these 'tokens' provide more evidence of the structured and entrenched regional divides. It is not so much as who we are (as perhaps the discussion of family name etc implies) but where we live and where we are born. As geographers, this will not be news to the authors of the paper but it feels like it deserves a more prominent line in the narrative of the paper. Indeed the compelling line stating that 'the message of this paper is that social mobility is historically and geographically contingent, hence unique' could do with more back up earlier on.

The work is of significance for all work on social mobility, as the authors point out, including in building the evidence base and empirically demonstrating how (dis)advantages afforded to people because of where they are born are then maintained between generations, and in showcasing the value of the 'tokens of geographic identity'. It is an original piece of work worthy of publication, employing robust methods applied to a novel dataset. The information provided are enough to support others in replicating the work with comparable data access.

RESPONSES TO THE REVIEWERS' COMMENTS

Reviewer #1 (Remarks to the Author):

Thank you for submitting this interesting & idea provoking piece of work (see below!).

>>> Thank you very much for the very positive comments.

I encourage the authors to take a step back from the whole and see whether a different series of subsections might just help the what we are doing, how and what we have found elements.

Rather than a standard Data & Methods and Results sections, I need to declare that I really do like, for successive subsections of work like this, that people say for each part what the resources were, what processing happened and what was found. This is important in work like this whereby one can forget what on earth it was the people did (in methods) to thereby understand the results. To a large extent you do this, but some better divisions and flagging / pinpointing the story would help.

Here, you are a little inconsistent. For example, the introduction moves into data source (the historic bits) but misses the consumer register which, as far as I can see, is not given sufficient explanation. Then, we get the measures being used (lines 111 onwards) and we move into some results while still in the intro.

I'd suggest the intro does what it should by having the social mobility elements and roughly then up to line 72. Then explain (if you can be encouraged to agree with me!) that there will be a set of analyses in which the sources, methods and findings will be self-contained. each of these then says what it will do.

>>> We have revised the paper to make it easier to follow and comprehend while following the standard sections specified by the Journal. We have restructured the Introduction by foregrounding work on family name origins/concentration, using the illustrative map. We have moved the equations to the opening part of Results section, also in light of the Editor's requirement to put technical details that are necessary to follow the main text at the beginning of the Results section.

In addition, we have now added brief descriptions and statistics about the historical censuses and the consumer registers in the Method section of the revised version.

Referees are often asked if something can be omitted and still have a good paper. I find the London subsection a bit of an outlier; not least because it doesn't fit with the GB coverage. I think this would standalone in another paper & maybe you could incorporate the mirror. i.e. names born elsewhere who end up in London.

>>> We appreciate your suggestion on a standalone London paper. However, after giving a careful consideration of your suggestion, we do prefer to retain the London subsection. This is because it closely corresponds with the geographic structure of future deprivation that is core to the paper's findings. We use historical life-time migration to London to demonstrate the impact of north-south population movement upon observed present-day IMD outcomes. We do agree with you that there are many other factors to be incorporated in future examination of the observed inequalities of outcome.

The geography of inter-generational social mobility in Britain

A few details:

* As noted, some detail about the consumer register

>>> A brief introduction to the consumer register and historical censuses has been added to the Method section.

* Better colour distinction in fig 1 (I can't see the light blue

>>> Thanks for raising this point. We have changed the colour schemes of the figures, including Figure 1, to make them clearer, consistent with the Journal style guidelines.

* It looks like the adjusted IMD is between 0 and 100 (and you say range standardisation somewhere later?)

>>> The IMD scores are percentiles ranging from 0 to 100 and we harmonised them between England/Wales and Scotland to make it a GB-wide index of neighbourhood conditions. We did not perform any range standardisation for individual UK country IMD scores. For better visualisation of spatial distributions of surnames with small numbers of bearers, we conducted range standardisation for the KDE contours, as stated in the caption of Figure 1.

* Can fig 2 have an everyone else's average (IQRs) to see how the families of interest vary from others?

>>> This is a very interesting point, and the common names Jones, Williams, Smith and Brown in the middle of Figure 2b give an approximate indication. But 'everyone' includes recent migrant family groups not present in significant numbers in the historical period – something that lies beyond the scope of this paper, but which we hope to address in future research.

A couple of ideas for future work:

What is the old and new geography of place name surnames?

What is the old and new geography re IMD of occupation names?

Good luck with all this which is really intriguing work!

Paul Norman

>>> These are very useful research ideas for future exploration. Thank you very much for your suggestions and overall comments.

Reviewer #2 (Remarks to the Author):

This paper uses the Britain-wide Index of Multiple Deprivation (IMD) from 2019/20 in combination with the 1851 census and the 2016 consumer register to examine social mobility over a period 170 years. Individuals are assigned to a “long established” family by their last name in 2016 and in 1851, assuming that individuals with the same (uncommon – nevertheless accounting for 63% of the population in 2016) last name stem from the same lineage. Using geo-references in the consumer index, the authors then calculate average deprivation scores for each family, predicted deprivation scores of each neighbourhood and future deprivation scores of parishes in 1851.

The geography of inter-generational social mobility in Britain

The method of using last names is not new and the authors fully acknowledge this. What is new, is the combination with population-wide modern data including geo-references which allows an examination of the geography in social mobility.

The method is sound and the results are novel and interesting. The main contribution, demonstrating the geography of social mobility, will be useful for other researchers but also has policy-relevant implications.

My main comment concerns how the results are put into perspective. The discussion of the main contribution of the paper could be more focused throughout the paper. This would not only improve readability but also bring out more clearly what we actually learn from this exercise (also in terms of context-related implications as opposed to data-related implications).

>>> We greatly appreciate your constructive reflections and feedback on our work. We have restructured aspects of our paper accordingly and focus more clearly on the original contributions of the work to geographical analysis.

The discussion on p. 11 line 384-407 summarizes the main contribution of this paper nicely: social processes exhibit geographical persistence. This is a very interesting and policy-relevant result. I would suggest the authors to focus on this and outline more clearly the implications of this demonstration. As it stands now, the authors discuss different motivations/implications not all of which can really be derived from the results. For example, the authors state several times throughout the paper (especially in the discussion) that the results demonstrate that it is not sufficient to look at social mobility over only two generations. This is true, however especially in recent years, with the availability of large historical datasets, the focus has already moved towards long(er)-run social mobility measures over several generations and the authors also cite the literature.

>>> Thank you for raising this recent shift towards longer term social mobility research. In the revised paper, we have added the indicative reference of Song et al. (2020). However, compared to research based upon probabilistic record linkage, use of occupation and income, or sample-based inter-generational studies, we do present our work as a new scale-free, population wide intergenerational approach.

X. Song *et al.*, Long-term decline in intergenerational mobility in the United States since the 1850s. *Proc. Natl. Acad. Sci. U. S. A.* 117(1):251-258 (2020).

Also, at times the paper is presented as showing how geography affects social mobility (e.g. p. 6 line 196: “Our exploratory analysis of the names.... Suggests the influence of distinctive local factors...”). We cannot derive this from the results, it may also be individuals/families exhibiting a certain (im)mobility pattern (or their wealth and land possessions – or lack thereof) leading to distinctive local factors. What we can derive is the persistence in the geographic pattern in social mobility – not the mechanism.

>>> We agree that the principal contribution is to demonstrate persistence in geographic patterns rather than specific social processes. We have rephrased this sentence in the revised version.

The analysis is done on 13,378 long established family groups. While these cover the majority of the population in 2019, their mobility patterns may be different from common namers, especially historically and in terms of geographical mobility. Also, populations are not actually “linked” but rather classified into family groups based on family names. As such, the statement that the analysis

The geography of inter-generational social mobility in Britain

is “truly intergenerational” (p.3 line 54) is somewhat overstated, also given the fact that the deprivation index is not an individual measure. This is unnecessary given the immense datawork of the paper.

>>> We think that we are candid in the paper about effects of common (and geographically widespread) names in the Discussion section. Common names such as Smith do not have apparent local or regional concentrations and their bearers’ circumstances mirror those of the underlying long-established population.

We use the word “couple” in the revised version to make absolutely clear that direct record linkage is not attempted. However we do not believe that the fact of coupling rather than direct linkage negates our claim that the work is “truly inter-generational”.

You are absolutely correct that the IMD is not a personal characteristic, but it does measure the mix of physical and social conditions in which any individual resides. As such it is, arguably, a less transient indicator of personal circumstances than income, occupation or other indicators used in generational studies.

The discussion includes a paragraph about immigration, especially from Ireland. The period after 1851 was, however, also a period of massive outmigration. Emigrants were on average poorer and less educated and emigrated in search of better opportunities. This also implies that those who stayed were those who deemed their opportunities at home sufficiently preferable to emigrating.

>>> These are valid and very interesting observations that we will investigate in our future research.

The illustrations (figures and maps) are very well chosen and illustrative. Especially figure 3, however, calls for a comparison between the modelled and the observed IMD. I understand the authors did not include the observed IMD as this is not the contribution of the paper, but it could be useful for the reader to see the difference between modelled and observed, for example.

>>> We have inserted a map of the observed IMD scores in the Supplementary Information. Thank you for this very useful suggestion.

Some more details on the methodology (especially the “linking” procedure, incl. match rates) and the different datasets (summary statistics, observation numbers, etc.) would be useful for readers who are interested in the supplementary material.

>>> Thank you. We have revised the Method section and included details of the data sources in the Method section accordingly.

Reviewer #3 (Remarks to the Author):

Thank you for the opportunity to review this paper, I very much like it. It takes advantage of digitised historical records in order to present the first analysis of inter-generational social mobility for the Victorian population in GB to the present-day population. Area-level deprivation is used to measure social mobility outcomes for 13,378 ‘long-established’ family groups, exploring how geography of ancestry relates to apparent differences in opportunity.

>>> Thank you very much for these very supportive words on our work.

The geography of inter-generational social mobility in Britain

The key finding emphasises the importance of context, and that – as some may argue is already known – ‘inequality is deep-seated, inter-generational and substantially geographic’. Though perhaps not as novel a finding as the paper feels like it implies, it is certainly a fresh and valuable set of evidence giving more substance to what is perhaps already articulated in wider sociological and economic literatures. What is most compelling about this paper is the arguments around under-use of ‘universal tokens of geographic identity’, and the way in which these ‘tokens’ provide more evidence of the structured and entrenched regional divides. It is not so much as who we are (as perhaps the discussion of family name etc implies) but where we live and where we are born. As geographers, this will not be news to the authors of the paper but it feels like it deserves a more prominent line in the narrative of the paper. Indeed the compelling line stating that ‘the message of this paper is that social mobility is historically and geographically contingent, hence unique’ could do with more back up earlier on.

>>> We very much appreciate these very useful reflections. We concur it is not a new idea that geography matters for understanding inter-generational social mobility. However, as you correctly point out, the main contribution of this work is the use of names as universal tokens and new micro level population data sources to provide population-wide empirical evidence that geography matters. This is different from conventional sample-based or occupation-based social mobility research. We have made this message more prominent in the revised version accordingly.

The work is of significance for all work on social mobility, as the authors point out, including in building the evidence base and empirically demonstrating how (dis)advantages afforded to people because of where they are born are then maintained between generations, and in showcasing the value of the ‘tokens of geographic identity’. It is an original piece of work worthy of publication, employing robust methods applied to a novel dataset. The information provided are enough to support others in replicating the work with comparable data access.

>>> We greatly appreciate your positive comments and suggestions.